# Off-Label Treatment for Severe Craniomaxillofacial Fractures in Low-Income Countries—A Novel Operation Method with the External Face Fixator

**DOI:** 10.3390/jcm11061488

**Published:** 2022-03-09

**Authors:** Christian Deininger, Valeska Hofmann, Marco Necchi, Susanne Deininger, Florian Wichlas

**Affiliations:** 1Department of Orthopedics and Traumatology, Paracelsus Medical University, Müllner Hauptstrasse 48, 5020 Salzburg, Austria; fwichlas@nolimitsurgery.com; 2No Limit Surgery, Ernest-Thun-Strasse 6, 5020 Salzburg, Austria; vhofmann@nolimitsurgery.com (V.H.); marco.necchi@hotmail.it (M.N.); susanne.deininger@mail.de (S.D.); 3Institute of Tendon and Bone Regeneration, Paracelsus Medical University, 5020 Salzburg, Austria; 4BG Trauma Centre, Department of Trauma and Reconstructive Surgery, University of Tübingen, 72076 Tübingen, Germany; 5Department of Surgery and Orthopaedics, Hospital Sterzing, Margarethenstraße 24, 39049 Sterzing, Italy; 6Department of Urology and Andrology, Paracelsus Medical University, Müllner Hauptstrasse 48, 5020 Salzburg, Austria

**Keywords:** low-income countries, external fixator, off-label treatment option, craniomaxillofacial fracture

## Abstract

Introduction: Craniomaxillofacial fractures (CMF) are common in low-income countries (LIC). Due to limited resources, treatment of these fractures usually consists of interdental wiring or immobilization with a Barton bandage to maintain the reduction by permanent occlusion. These non-surgical treatment methods often lead to unsatisfactory results, such as a disturbed dental occlusion and lockjaw. The aim of this study is to present an off-label treatment option for CMF by applying a hand fixator as external face fixator (EFF) and to demonstrate the surgical method in detail. Materials and Methods: The feasibility and postoperative outcomes of this new off-label operation technique were evaluated by analyzing patients with CMF (*n* = 13) treated at an NGO hospital in Sierra Leone between 2015 and 2019. Results: The application of the EFF was feasible. The biggest advantage compared to the conventional non-operative methods was, that a dynamic occlusion was still possible during the 6 weeks healing period. Hence, patients could eat and drink almost normally and perform dental hygiene with the EFF in place. We did not discover pintrack infections or other complications. Three patients developed an oronasal fistula due to traumatic a palatal bone loss of about 7–8 mm which was treated by a palatal mucoperiosteal flap 15–20 days after the first operation. Discussion and Conclusions: In LIC, where plate osteosynthesis for CMF cannot be performed due to limited resources the application of an EFF is a promising alternative for a better outcome and an improved quality of life for the patients.

## 1. Introduction

Craniomaxillofacial fractures (CMF) are common in low-income countries (LIC), due to the lack of passive safety systems, such as helmets and airbags in road traffic [1]. After road traffic accidents, besides severe fractures of the extremities, CMF pose a special problem for treatment [2,3]. The lack of computed tomography (CT) scanners and suitable osteosynthesis, mostly mini-plates, makes the fixation of these fractures difficult [4]. In LIC, the so-called “Joe Hall Morris fixation”, an airway tube filled with bone cement, can be used as an alternative for external fixators in the treatment of mandible fractures [5]. External fixators designed for small bones, i.e., hand and wrist fractures, are also well established for temporary fixation in mandible fractures [6,7]. The use of an Ilizarov-type external fixator for CMF was first described in a case report by Hirara et al. in 2019. This type of fixation was chosen as an alternative to plate osteosynthesis due to an uncontrollable intraoral bleeding caused by a severe panfacial fracture of a 70-year-old woman [8]. In high-income countries (HIC) these factures are treated with mini-plates, even in extensive comminuted fractures [9,10].

Complex reconstruction of CMF can be performed after thorough planning only. This planning can be achieved by CT scans and 3D reconstruction. In LIC, CT and 3D reconstruction are often not accessible [11]. Without CT scans, the use of plates for complex reconstruction is impossible as approaches must be planned, and instabilities defined. The reconstruction of CMF becomes cumbersome when CT scans and mini-plates are not available [12].

External fixators are the implant for austere environments. Different types of external fixators can be used for small bones, such as hand and foot fractures, and for long bones and pelvis [13]. As external fixators are usually the first implant for emergency hospitals in trauma care, their use for CMF becomes evident. We widened the indication for the use of hand external fixator for the CMF as an external face fixator (EFF) in a LIC setting [8,14]. This model was chosen because, on the one hand, it is also usually available in LIC and the 4 mm diameter of the pins seems clinically adequate. In addition, the preclinical and early clinical management of patients with complex midface fractures is already a problem in many cases. Securing the airway can be complicated by sometimes severe bleeding or swelling of the soft tissues [15].

The goal of this study was to evaluate the definitive treatment of CMF with the EFF in the LIC Sierra Leone. We analyzed the surgical feasibility of EFF for CMF, the rate of complications and infections and the postoperative occlusion and the rate of lockjaws. The results of this work could generate a solution to fractures otherwise left untreated or treated non-operatively. Non-operative treatment of these fractures always means strict immobilization of the maxilla and mandible to maintain occlusion and thus reduction. Accordingly, these patients cannot eat solid food for 6 weeks, cannot speak adequately, dental hygiene is limited, and the risk of lockjaw is high. We firstly present a flowchart on the clinical management of CMF in LIC using the EFF from admission to discharge of the patients and secondly a recommended course of action for intraoperative clinical localization of the instability, for placement of the EFF pins, and the reduction by temporary, intra-operative occlusion.

## 2. Materials and Methods

### 2.1. Patients

From 2015 to 2019, 13 patients with CMF and three with isolated mandible fractures were treated with the EFF in a non-governmental organization (NGO) hospital in Freetown, Sierra Leone, Africa. Ten CMF patients had an additional mandible fracture (10/13).

Two patients were female, 14 were male. We applied 13 EFF to CMF and 13 on the mandible. The cause of fracture was motorcyclist road traffic accident without helmet in all cases. All fractures were open (*n* = 16), 81.25% (*n* = 13) had lacerations in the nasal or frontal region, 43.75% (*n* = 7) had at least intraoral wounds. Nine patients with CMF were not treated by the EFF due to a lack of implants or operative capacities. They were not included.

### 2.2. Setting

The NGO hospital is equipped with 85 beds, eight intensive care beds without ventilator, three operation theaters (OT), an outpatient department, a room for casting/splinting, and one for physiotherapy. The hospital’s admission criteria were trauma victims, patients requiring general surgery and pediatric patients [11].

### 2.3. Diagnostics

The CMF was diagnosed by clinical examination and conventional X-ray imaging. The clinical signs of fracture were the presence of pain and swollen soft tissues, combined with the reported trauma mechanism and bony instability tested by pulling on the upper jaw. The bony landmarks of the facial skull were palpated. The X-ray planes were anteroposterior and lateral (Figure 1A,B). A CT scan was not available.

### 2.4. Treatment

The decision-making algorithm for patient guidance is described in Figure 2. The EFF used for CMF was a fixator designed for distal radius fractures and fixation of the radial shaft to the metacarpal bones (Hoffmann II compact external, Stryker Trauma AG, Selzbach, Switzerland) [16,17]. The pin diameter was 4 mm. The first use of the EFF for CMF was done after the reconstruction of an open frontal sinus fracture with a periostal flap. In this case, the fixation was necessary for flap protection. First, stab incisions following the Langerhans lines were made. An electric power drill was used, and the pin was inserted with a T-chuck. In this case, the first two pins were inserted into the zygomatic bone. The second two pins were inserted into the maxilla above the dental root level. Here, the direction of the surgeon’s left finger inserted into the mouth laying with the tip on the palate of the patient guided the way. The insertion of the pin was stopped when the tip of the Schanz Screw was perceived under the palatal mucosa. As instability between the maxilla/zygomatic bone and the frontal bone persisted the fixation was extended supraorbitally. The clinically defined safe zones for pin insertion can be seen in Figure 3.

In 11 of the 13 CMF (84.62%), the zygomatic bone was comminuted, and a stable pin insertion was impossible. In these cases, the first pin was inserted into the Maxilla just above the dental roots below the nostrils, one on each side. In the case of a palatine split it was reduced first under vision and by palpation of the fracture. In addition, the reduction of the upper row of teeth was controlled. Afterwards the second two pins were inserted slightly lateral to the first ones in the maxilla, again close to the dental roots in the hard palate. These pins were necessary because only one pin on each side was clinically deemed too unstable. The next pins were inserted into the frontal bone close to the orbit in the lateral orbital third and connected to the first ones (Figure 1C,D and Figure 4). The pin location was determined clinically referring to anatomical landmarks [18]. The reduction always was set up by temporary occlusion. The operation was performed without the use of an image intensifier.

All mandible fractures were treated with external fixation [17]. First, the pins were placed in safe zones of the mandible as according to guidelines of the “Arbeitsgemeinschaft für Osteosynthesefragen” (AO) [18]. Fracture reduction was performed after debridement of the gingiva and anatomic reduction. Then, the occlusion was controlled, and the frame fixed.

When the midface and the mandible where fractured at the same time, the mandible was reduced and fixed first. The reduction of the maxilla was controlled by using the reduced mandible as a template and testing the occlusion. The EFF and the external fixator for the mandible fractures are two separate osteosyntheses. The reduction and fixation were always performed bottom up.

### 2.5. Aftercare

The EFF for CMF was left in situ for 6 weeks and the patients were allowed to drink and eat normally, avoiding hard food. After removal of the EFF, open wound treatment of the incisions was performed. The management of patients with CMF can be found as a flowchart in Figure 2.

We evaluated the EFF as new operation method for its feasibility and analyzed difficulties in the application of the fixator and fracture reduction. Furthermore, the required diagnostics, the clinical course, and outcome of the patients are described. In addition, we have listed possible complications and pitfalls.

## 3. Results

### 3.1. The CMF

The application and handling of the EFF for CMF was feasible. The pin dimension of 4 mm diameter was appropriate and provided a good purchase in the maxilla, zygomatic, and frontal bone. The advantage of this pin size was that it was almost always available. If 3.5 mm pins were also in stock, these were used for the palatal bone. They felt clinically more appropriate in this region. An image intensifier was not needed during the operation. The reduction could be performed using the occlusion as reference [19]. This was difficult in patients who were oropharyngeally intubated, as the tube hindered clinical control of a correct occlusion. In these cases, clinical control of occlusion after extubation and, if necessary, improvement of reduction in OT under sedoanalgesia were performed as standard of care. The operative procedure showed no adverse events, and the postoperative course was uneventful. Due to the retained dynamic occlusion the patients could eat and drink (Figure 1C,D). Their occlusion after removal of the EFF was clinically restored without signs of lockjaw. No complications were detected during the 6 weeks period of convalescence (0/13). Clinically all fractures were stable and had healed (Figure 1E,F).

We saw no pintract infection after treatment and the wounds intraorally had all healed without complications. There were no peripheral neurological lesions in the head and face. Open hard palate fractures had healed and were covered with gingiva in most of the cases (10/13, 76.92%).

### 3.2. Soft Tissue

Three (23.08%) patients with intraoral wounds and a palatal bone loss of about 7–8 mm developed an oronasal fistula with passage of water and food in the nasal cavity. These were closed within 15–20 days after primary surgery by a palatal mucoperiosteal flap [20]. All three healed without further incidents.

### 3.3. Mandible

The application of the external fixation on the mandible was feasible. The pin dimension of 4.0 mm was appropriate and showed a good purchase in the bone. One fracture was on two levels and needed six pins, two per fragment. The placement in the ramus of the mandible was cumbersome as the soft tissues permitted less control of the fragment but could be achieved [17].

When the midface and the mandible were fractured at the same time, the fixation could be performed as described without further noticeable problems.

### 3.4. Damage Control Surgery

Five (38.46%) patients needed an urgent tracheostomy due to rapid desaturation on arrival. The cause was either intraoral hemorrhage and soft tissue swelling (4/5), or a severely dislocated mandible fracture with the tongue being displaced posteriorly (1/5). In these cases, the EFF was performed immediately. Due to the tracheostomy, the occlusion could be performed properly. Hence, no adjustment of the fixator was necessary.

In 6 (46.15%) cases the swelling was increasing, but the airway still open. Those patients have been oropharyngeally intubated and operated the same day. Here, the tube hindered the testing of an adequate occlusion during the operation. After 5 days, there was usually a significant improvement of swelling of the soft tissues and the general condition of the patients. After extubation, the occlusion was clinically checked. In six (75%) patients the reduction was corrected under sedoanalgesia with spontaneous breathing in the OT. No pins were changed, but clamps and rods were readjusted.

In two (15.38%) patients the airway was not in danger and the EFF was performed as planned surgery the next days.

### 3.5. Diagnostics

In HIC, the surgical treatment of CMF requires a precise planning. For this, a CT scan of the midface is performed as standard. This allows the fracture morphology to be analyzed preoperatively and the operation to be planned in detail [9]. In LIC, the fracture diagnostic by X-ray and clinical examination was limited. It was possible to determine the presence of an unstable CMF but not the exact fracture pattern.

The diagnostics of mandible fractures was feasible and comparably easy. Clinical examination was the determining method of investigation. The X-ray in two planes was a supplement to the clinical examination.

## 4. Discussion

The treatment of CMF in LIC is difficult due to the lack of CT scans and mini plates [9].

Prior to the introduction of the EFF, this hospital used a non-operative approach or an interdental wiring when possible. A functional treatment of CMF cannot be provided with interdental or intermaxillary wiring, especially in comminuted complex fractures. A K-wire fixation from one zygomatic arch to the other is amendable for some specific pathologies only, Lefort II fractures for instance, but should be secured by additional fixation of occlusion [21,22]. The non-operative treatment of these fractures consists of reducing the occlusion and fixing it with a Barton bandage [23]. The majority of these treatments have in common that they immobilize the jaw for the duration of bony healing and hence do not allow a dynamic occlusion. This non-surgical treatment regularly leads to unsatisfactory results. Patients suffer during conservative therapy with the Barton Bandage. The outcome is correspondingly poor. The bones heal poorly reduced, if at all. Normal occlusion is no longer possible in many cases. Eating and speaking become a challenge.

The external hand fixator seems to be the ideal solution for these CMFs as it allows the targeted fixation of specific fractured regions in CMF and the functional aftercare.

The challenge in treating CMF in LIC is to analyze the fracture pattern and to determine the level of instability. The use of X-ray imaging is limited and often insufficient in diagnosing CMF. Due to the lack of CT scanners, the detection of the level of instability can be assessed by clinical examination only. This is performed by pulling the upper jaw and fixing manually the skull on the zygomatic arch or supraorbitally.

The main advantage, besides the probable better healing in terms of reduction and bony union, is the patient’s comfort and quality of life. With exception of hard food, the patient was able to eat and drink normally, dental hygiene was hardly limited, and the patients could talk. The application of the EFF does not require specialized orthopedic trauma knowledge and is technically feasible for most surgeons, although the safe zones for pin placement should be known. Figure 3 gives an overview over the used pin areas [18]. The insertion of the second palatine pins is difficult, because the dental roots of the upper teeth are close to the fracture line. In our experience, you need two maxillary pins per side to achieve an adequate stability. The pins are either close to the dental roots or to the fracture. Our proposal is to place the first pins in the hard palate right beneath the nostrils and the second palatal pins slightly lateral to them. Hihara et al. showed similar insertion site selection in their case study published in 2019. In particular, the pins in the hard palate were placed in the same location [8]. The alternative is to insert one pin in the zygomatic bone on each side. However, it was broken in over 80% of our patients and therefore not available for pin insertion. In Hihara’s case he could use the Zygomatic bone for pin insertion [8].

The off-label use of these external hand fixators was born out of necessity to treat the CMF because the application of interdental wires was not possible [17]. The patients had incomplete denture or conical shaped teeth that made it impossible to fix the wires as they slipped off the teeth [24]. In addition, the patients were regularly missing their front teeth due to the accident. As plates, screws, and K-wires did not seem appropriate for fixation, external hand fixators were applied and proved their usefulness. However, even external fixators are not always available, especially hand fixators, or should be reserved for urgent fractures.

In these cases, the use of the Joe-Hall-Morris technique, an airway tube filled with bone cement to connect the pins could be a solution [25]. Another possibility would be to use less pins to fix the face. Especially in typical LF III fractures, a diagonal frame would make sense to spare the number of pins, as all resources are limited in LIC. For emergency procedures, the buccal occlusion can be controlled and a fixation from the mandible to supraorbital bone performed as already described. In some circumstances even an application under local anesthesia might be feasible.

In a study published in 2010, Cienfuegos et al. demonstrated the treatment of palatal fractures with an angle stable 2.0 mm locking plate system as external fixation in 45 patients. The conclusion of his group was that this type of treatment is an excellent alternative to internal fixation [26]. This treatment option is another alternative for palatal fractures, provided the plate instrumentation is available. A combination with the EFF would be conceivable.

In HIC this fixation method might be suitable for temporary fixation in severely injured patients [13]. After the treatment of life-threatening injuries, the external fixation of CMF could be performed on intensive care unit (ICU) or during second look surgeries. Three-dimensional reconstruction of polytrauma CT scans of the face could be reconstructed and instabilities of the face addressed. The ICU care could be easier until definitive plating of the CMF.

The superiority of the external fixation compared to non-operative treatment or inter maxillary wiring needs to be proved by further studies. The lack of complications in this collective combined with the practicability of the surgery and the comfort of the patient are arguments for its use. Untreated CMF have a risk for non-union. When the fractures are not reduced adequately, the callus formation leads to facial asymmetries. Non corrected reduction in the ocular region can lead to enophthalmos and diplopia, and non-reduced occlusion leads to malocclusion [27].

The treatment algorithm of CMF foresees interdental fixation and reduction of the mandible to the maxilla. The interdental ligatures permit the reduction of occlusion using the stabilized mandible as template. Some techniques describe a wiring of the mandible to the zygomatic arch intraorally. However, these techniques require an unfractured mandible and zygomatic arch, and as most CMF present with a concomitant fracture of the mandible (10 out of 13) (76.92%), the reduction and fixation of both by wiring might be difficult or impossible [28]. Even when achieved, the occlusion must be fixed for the duration of healing and the mouth closed. Conclusively, the interdental fixation seems to be an unideal compromise born out of the lack of other solutions and cannot address the pathology of CMF directly. Nevertheless, the use of external fixators for CMF requires critical analysis and planning. The use of pin diameter and frame, as well as ideal pin placement, need to be researched and determined.

### Limitations of the Study

The described surgical technique was only clinically investigated for its feasibility and outcome. The exact location of the pins, the strength of the pins in the bone, and possible other “safe zones” will be investigated in a follow-up project on the cadaver.

## 5. Conclusions

In LIC, resources are limited and severe injuries to the facial skull are comparatively as common [29,30]. As a result, the treatment of CMF is challenging [31]. Until now, such fractures were treated non-operatively with a Barton Bandage or interdental wiring [28]. This resulted in insufficient fracture reduction, lockjaw, and a significant reduced quality of life.

Dynamic occlusion was possible with the EFF in situ, and after removal. After completed treatment all fractures were clinically consolidated. The correct tooth occlusion was restored.

The novel surgical method represents an alternative treatment option of CMF in LIC.

## Figures and Tables

**Figure 1 jcm-11-01488-f001:**
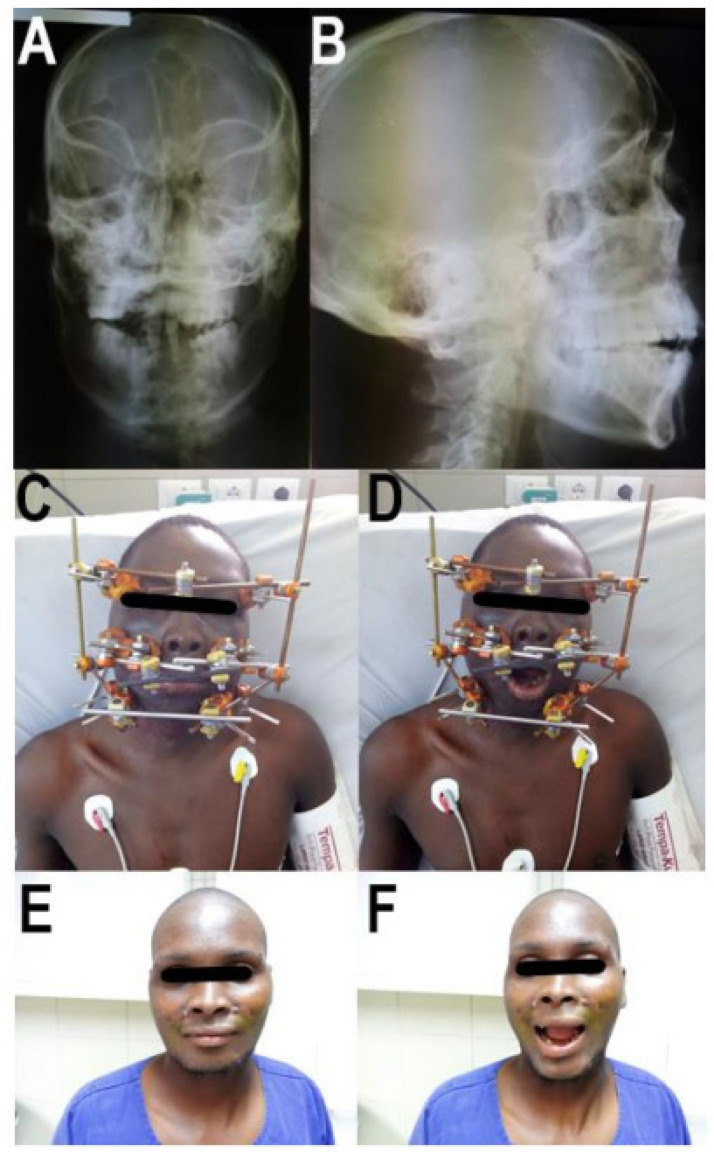
Case report of a patient with craniomaxillofacial fracture (CMF), treated with the External Face Fixator (EFF). Pictures (**A**,**B**) show preoperative radiographs of the skull in anteroposterior and lateral planes. Although a significant instability caused by a CMF is clinically diagnosed intraoperatively, the full extent is difficult to determine in the X-ray. Pictures (**C**,**D**) show the patient with the EFF in place and the range of motion of the temporomandibular joint during dynamic occlusion. Pictures (**E**,**F**) document the final result after removal of the fixator and fracture healing 8 weeks after the operation.

**Figure 2 jcm-11-01488-f002:**
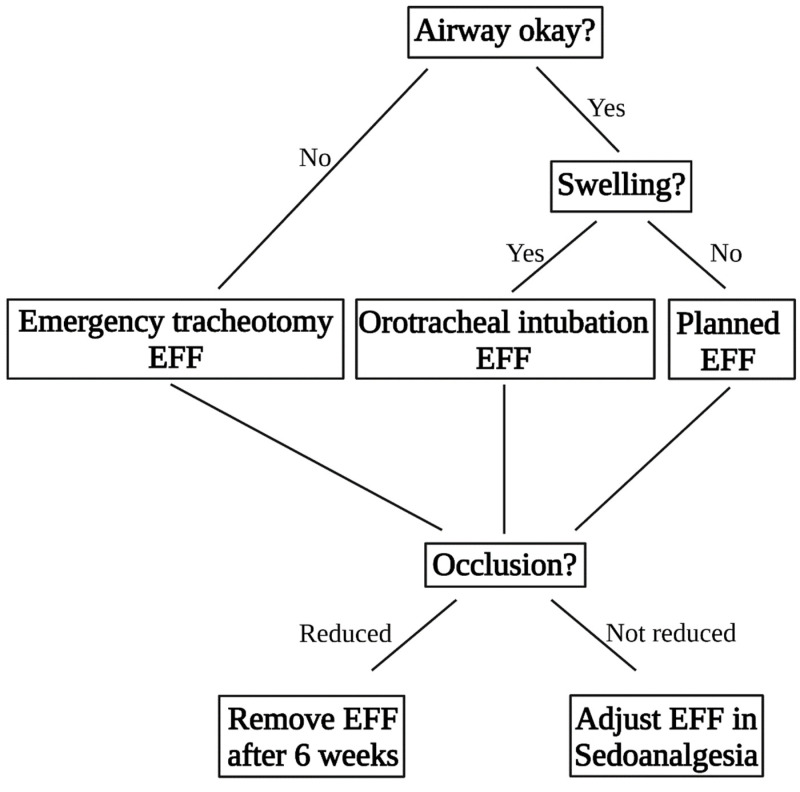
Flowchart and decision-making of the patient guidance from the first contact to the operation to healing.

**Figure 3 jcm-11-01488-f003:**
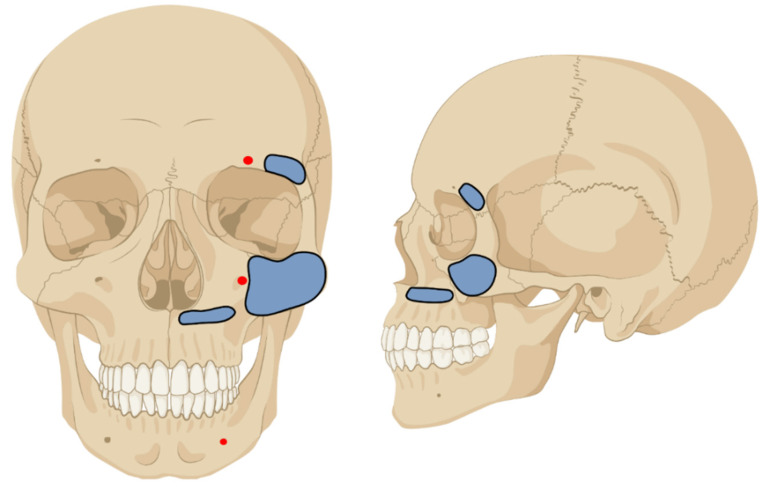
Nerve exit points and clinically tested safe zones for pin insertion for the External Face Fixator. Created with BioRender.com.

**Figure 4 jcm-11-01488-f004:**
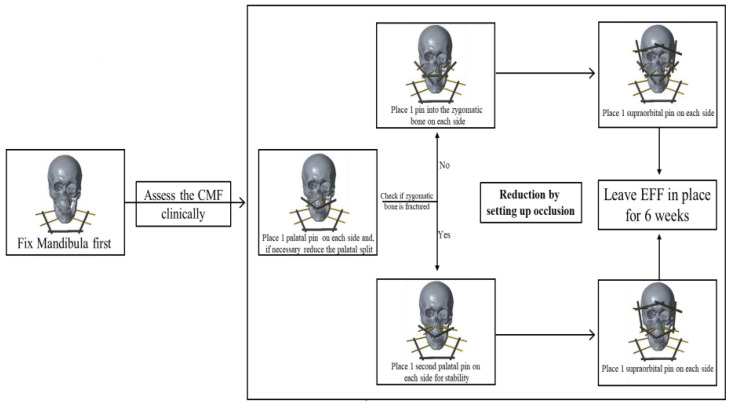
Flowchart and decision-making of the operation technique of the External Face Fixator (EFF) for Craniomaxillofacial fractures (CMF).

## Data Availability

Data can be obtained from the authors on request.

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
