# Peer review of "Off-Label Treatment for Severe Craniomaxillofacial Fractures in Low-Income Countries—A Novel Operation Method with the External Face Fixator"

_jcm, 2022, doi:10.3390/jcm11061488_

Round 1
Reviewer 1 Report
it is a nice study and the authors should be complimented.
my advice is to use for maxillary fixation the thick bone buttresses as described in many papers , the nasomaxillary buttress and the zygomatic-maxillary buttress , also in the lower part , taking into account the presence of molar roots. the authors should clarify if this method was used or could be used to reduce the concomitant fracture of the zygoma.
Author Response
Dear Reviewer,
thank you very much for your comments. Please find attached our answers.
my advice is to use for maxillary fixation the thick bone buttresses as described in many papers , the nasomaxillary buttress and the zygomatic-maxillary buttress , also in the lower part , taking into account the presence of molar roots. the authors should clarify if this method was used or could be used to reduce the concomitant fracture of the zygoma.
Thanks for the comments and question. In general, the molar routes were not a problem because the dental status, even in the relatively young patient in Sierra Leone, is not comparable to that in a developed country. When the zygomatic bone was fractured, which was usually the case, the main approach was to achieve a perfect reduction of the dental realm by occlusion. The construct was stable mainly due to the pins just below the nostrils and the suprorbital pins. In the case of comminuted fractures in the region of the zygomatic bone, this was not treated by a pin.

Reviewer 2 Report
there is no novelty in this paper, external pin fixation known since long and documented facts about the external fixation such as The apparatus can be bulky and can have a negative aesthetic appearance during the treatment period that can last several weeks to several months depending on the clinical situation.. this would definitely affect the quality of life of an individual.
The author mentioned in line 38 in the abstract their technique affects the quality of life ...this conclusion is based on what the author did not assess the patient quality of life by and objective ways.
The author mentioned that there is no complication was encounter in his samples. However, Ellis et al found 23.5% of patients treated with ex-fix developed nonunion requiring bone grafts, while 11.7% of patients treated with ex-fix developed malocclusion (an overall complication rate of 35%). Rates of nonunion and malocclusion were much higher with ex-fix treatment than with rigid internal fixation
The author is not clear about sample size in material section, line 94 mentioned “All fractures were open, 81.25% (n=13) had lacerations in the nasal or frontal 93 region, 43.75% (n=7) had at least intraoral wounds. Nine patients with CMF were not 94 treated by the EFF due to a lack of implants or operative capacities. They were not included”.
Author Response
Dear Reviewer,
thank you very much for your comments. Please find attached our answers.
there is no novelty in this paper, external pin fixation known since long and documented facts about the external fixation such as The apparatus can be bulky and can have a negative aesthetic appearance during the treatment period that can last several weeks to several months depending on the clinical situation.. this would definitely affect the quality of life of an individual.
The author mentioned in line 38 in the abstract their technique affects the quality of life ...this conclusion is based on what the author did not assess the patient quality of life by and objective ways.
Thank you for your comments. Quality of life was indeed not evaluated by means of a standardized questionnaire. This is mainly due to the clinical setting in Sierra Leone. With a poverty rate of 56.8% in 2022, Sierra Leone ranks 13th according to a report by: Poverty Rate by Country 2022 (worldpopulationreview.com). Here, the focus is on providing the most adequate care possible to injured patients. Thus, the quality of life was evaluated only subjectively by the surgeons on site and compared their experience with the alternative treatment methods (Barton Bandage, interdental wiring). In regions of the third World, restoration of the bony facial skull and prevention of permanent nasopharyngeal fistula is the primary focus. No complaints regarding aesthetics were expressed by patients treated with the External Face Fixator (EFF) during the 6-week treatment period. The Barton Bandage as an alternative showed significantly greater impairment and decreased compliance due to the rigid occlusion.
The author mentioned that there is no complication was encounter in his samples. However, Ellis et al found 23.5% of patients treated with ex-fix developed nonunion requiring bone grafts, while 11.7% of patients treated with ex-fix developed malocclusion (an overall complication rate of 35%). Rates of nonunion and malocclusion were much higher with ex-fix treatment than with rigid internal fixation
The aim of our study is to show the superiority of the EFF compared to other methods available in low-income countries. Treatment of complex midface fractures with angle stable miniplates is usually not possible here. On the one hand, because no computer tomograph is available for pre-operative planning and, on the other hand, because the required surgical equipment is also lacking. With the EEF and angle stable stabilization of complex fractures is possible, and a nasopharyngeal fistula can also be reduced and fixed. This is difficult or impossible with the alternatives Barton bandage and interdental wiring.
The rate of actual pseudarthrosis could not be analyzed due to the lack of a CT scanner. These data would certainly be interesting. Clinically, the follow-up 2 weeks after removal of the EFF showed good occlusion in all cases and satisfied patients. In our cases there were no clinically signs of pseudarthrosis. But this is of course hard to examine without ac CT scan.
The author is not clear about sample size in material section, line 94 mentioned “All fractures were open, 81.25% (n=13) had lacerations in the nasal or frontal 93 region, 43.75% (n=7) had at least intraoral wounds. Nine patients with CMF were not 94 treated by the EFF due to a lack of implants or operative capacities. They were not included”.
A total of 16 patients were included. 13 had complex midface fracture (CMF), 10 had CMF with a mandibular fracture. 3 patients had an isolated CMF and another 3 patients had an isolated mandibular fracture.
All fractures were open fractures (16). Either soft tissue injuries were evident in the facial region (13), intraoral (7), or both.
Reviewer 3 Report
Dear authors,
the article is interesting for practitioners in the field.
MAJOR:
Abstract
This method is not mentioned as an off label treatment.
What is A.N.O.M.W.T.E.F.?
Material and method
- Study design is not very clear. Was any comparison with another method?
- Ethical approval before the study should be mentioned.
- Inclusion and exclusion criteria are not clear. I wonder, for instance, if diabetic patients, osteoporotic were included or not.
Results
- there are no demographic data (no mentions about age, other associated diseases...)
Author Response
Dear Reviewer,
thank you very much for your comments. Please find attached our answers.
Dear authors,
the article is interesting for practitioners in the field.
MAJOR:
Abstract
This method is not mentioned as an off label treatment.
Thanks, we included this in the manuscript. Line: 24 and 27.
What is A.N.O.M.W.T.E.F.?
It is an abbreviation for “A new operation method with the external face fixator” used by the journal
Material and method
- Study design is not very clear. Was any comparison with another method?
A direct comparison with another method available in a low-income country (Barton bandage or interdental wiring) was not possible, as the External Face Fixator has almost completely replaced these techniques in this hospital. Therefore, a comparison can only be made subjectively and purely clinically. Here, however, the superiority of the new method was shown.
- Ethical approval before the study should be mentioned.
Line 345: Ethics approval: An ethics vote has been obtained from the local ethical committee. (Number: 1198/2021)
- Inclusion and exclusion criteria are not clear. I wonder, for instance, if diabetic patients, osteoporotic were included or not.
Results
- there are no demographic data (no mentions about age, other associated diseases...)
Included were all complex midface fractures treated at this hospital and a hand fixator set was available. Side diagnoses were not recorded due to the setting. In general, the patients were male patients (14/16) in their early 20s who were treated after a motorcycle accident.
Round 2
Reviewer 2 Report
There is no novelty in this paper, external pin fixation is known for long and documented facts about the external fixation such as The apparatus can be bulky and can have a negative aesthetic appearance during the treatment period that can last several weeks to several months depending on the clinical situation.. this would definitely affect the quality of life of an individual.
The author mentioned in line 38 in the abstract their technique IMPROVED the quality of life ...this conclusion is based on what the author did not assess the patient quality of life by ON objective ways.
The author mentioned that there is no complication was encountered in his samples. However, Ellis et al found 23.5% of patients treated with ex-fix developed nonunion requiring bone grafts, while 11.7% of patients treated with ex-fix developed malocclusion (an overall complication rate of 35%). Rates of nonunion and malocclusion were much higher with ex-fix treatment than with rigid internal fixation
The author is not clear about sample size in the material section, line 94 mentioned “All fractures were open, 81.25% (n=13) had lacerations in the nasal or frontal. LINE 93 region, 43.75% (n=7) had at least intraoral wounds. Nine patients with CMF were not treated by the EFF due to a lack of implants or operative capacities. They were not included”.
Finally, the authors have mentioned 3 patients developed an oroantral fistula due to traumatic palatal bone loss .. is this surgical trauma or from RTA. If it is from the accident why surgeon did not manage them on the same operation
Author Response
Dear Reviewer,
Thank you very much for your comments and remarks. We have adjusted the manuscript and answered the questions in the attached document.
Yours sincerely
The authors
there is no novelty in this paper, external pin fixation known since long and documented facts about the external fixation such as The apparatus can be bulky and can have a negative aesthetic appearance during the treatment period that can last several weeks to several months depending on the clinical situation.. this would definitely affect the quality of life of an individual.
Thank you very much for your comment. Could you share the literature where you found a similar technique? We have only one case report in our research: "A novel fixation method for panfacial fracture using an Ilizarov-type external fixator by Hihara, M. et al, 2019".
The author mentioned in line 38 in the abstract their technique affects the quality of life ...this conclusion is based on what the author did not assess the patient quality of life by and objective ways.
Thank you for your comments. Quality of life was indeed not evaluated by means of a standardized questionnaire. This is mainly due to the clinical setting in Sierra Leone. With a poverty rate of 56.8% in 2022, Sierra Leone ranks 13th according to a report by: Poverty Rate by Country 2022 (worldpopulationreview.com). Here, the focus is on providing the most adequate care possible to injured patients. Thus, the quality of life was evaluated only subjectively by the surgeons on site and compared their experience with the alternative treatment methods (Barton Bandage, interdental wiring). In regions of the third World, restoration of the bony facial skull and prevention of permanent nasopharyngeal fistula is the primary focus. No complaints regarding aesthetics were expressed by patients treated with the External Face Fixator (EFF) during the 6-week treatment period. The Barton Bandage as an alternative showed significantly greater impairment and decreased compliance due to the rigid occlusion.
The author mentioned that there is no complication was encounter in his samples. However, Ellis et al found 23.5% of patients treated with ex-fix developed nonunion requiring bone grafts, while 11.7% of patients treated with ex-fix developed malocclusion (an overall complication rate of 35%). Rates of nonunion and malocclusion were much higher with ex-fix treatment than with rigid internal fixation
The aim of our study is to show the superiority of the EFF compared to other methods available in low-income countries. Treatment of complex midface fractures with angle stable miniplates is usually not possible here. On the one hand, because no computer tomograph is available for pre-operative planning and, on the other hand, because the required surgical equipment is also lacking. With the EEF and angle stable stabilization of complex fractures is possible, and a nasopharyngeal fistula can also be reduced and fixed. This is difficult or impossible with the alternatives Barton bandage and interdental wiring.
The rate of actual pseudarthrosis could not be analyzed due to the lack of a CT scanner. These data would certainly be interesting. Clinically, the follow-up 2 weeks after removal of the EFF showed good occlusion in all cases and satisfied patients. In our cases there were no clinically signs of pseudarthrosis. But this is of course hard to examine without ac CT scan.
The author is not clear about sample size in material section, line 94 mentioned “All fractures were open, 81.25% (n=13) had lacerations in the nasal or frontal 93 region, 43.75% (n=7) had at least intraoral wounds. Nine patients with CMF were not 94 treated by the EFF due to a lack of implants or operative capacities. They were not included”.
A total of 16 patients were included. 13 had complex midface fracture (CMF), 10 had CMF with a mandibular fracture. 3 patients had an isolated CMF and another 3 patients had an isolated mandibular fracture.
All fractures were open fractures (16). Either soft tissue injuries were evident in the facial region (13), intraoral (7), or both.

Reviewer 3 Report
Dear authors,
you did not give complete answers to all the points I raised up.
Author Response
Dear Reviewer,
Thank you very much for your comments. We have added the missing answers.
With kind regards
The authors
Dear authors,
the article is interesting for practitioners in the field.
MAJOR:
Abstract
This method is not mentioned as an off label treatment.
Thanks, we included this in the manuscript. Line: 23 - 25 and 27 – 29..
- The aim of this study is to present an off-label novel treatment option for CMF by applying a hand fixator as external face fixator (EFF) and to demonstrate the surgical method in detail.
- The feasibility and postoperative outcomes of this new off-label operation technique were eval-uated by analyzing patients with CMF (N=13) treated at an NGO hospital in Sierra Leone between 2015 - 2019.
What is A.N.O.M.W.T.E.F.?
It is an abbreviation for “A new operation method with the external face fixator” used and created by the journal.
Material and method
- Study design is not very clear. Was any comparison with another method?
A direct comparison with another method available in a low-income country (Barton bandage or interdental wiring) was not possible, as the External Face Fixator has almost completely replaced these techniques in this hospital. Therefore, a comparison can only be made subjectively and purely clinically. Here, however, the superiority of the new method was shown. A randomized trial, as necessary in the Western world, is thus very difficult to conduct in a low-income country. One limitation of our study is the lack of a comparison group. However, the possibility of food intake and oral hygiene, as well as the increased compliance of the patients foster the application of the EFF. Our study group has over 15 years of experience as trauma surgeons in Africa.One colleague in our group has lived and worked exclusively in third world countries for over 10 years.
- Ethical approval before the study should be mentioned.
Line 345: Ethics approval: An ethics vote has been obtained from the local ethical committee. (Number: 1198/2021)
- Inclusion and exclusion criteria are not clear. I wonder, for instance, if diabetic patients, osteoporotic were included or not.
Due to the very rudimentary equipment of the hospital and the absolute emergency medicine, secondary diseases such as diabetes mellitus, hypertension or similar were not recorded. However, osteoporosis seems unlikely because the patients were generally in their early 20s.
Results
- there are no demographic data (no mentions about age, other associated diseases...)
Included were all complex midface fractures treated at this hospital and a hand fixator set was available. Side diagnoses were not recorded due to the setting. In general, the patients were male patients (14/16) in their early 20s who were treated after a motorcycle accident.